# Antibody-induced internalisation of retroviral envelope glycoproteins is a signal initiation event

Veera Panova[1], Jan Attig[1], George R. Young[2], Jonathan P. Stoye[2,3], George Kassiotis[1,3]*

1 Retroviral Immunology, The Francis Crick Institute, United Kingdom, 2 Retrovirus-Host Interactions, The Francis Crick Institute, London, United Kingdom, 3 Department of Medicine, Faculty of Medicine, Imperial College London, London, United Kingdom

* george.kassiotis@crick.ac.uk

**Data Availability Statement:** Data were deposited at the EMBL-EBI repository (www.ebi.ac.uk/arrayexpress) under accession numbers ERP120018 (EL4 cells; https://www.ncbi.nlm.nih.

## Abstract

As obligate parasites, viruses highjack, modify and repurpose the cellular machinery for their own replication. Viral proteins have, therefore, evolved biological functions, such as signalling potential, that alter host cell physiology in ways that are still incompletely understood. Retroviral envelope glycoproteins interact with several host proteins, extracellularly with their cellular receptor and anti-envelope antibodies, and intracellularly with proteins of the cytoskeleton or sorting, endocytosis and recirculation pathways. Here, we examined the impact of endogenous retroviral envelope glycoprotein expression and interaction with host proteins, particularly antibodies, on the cell, independently of retroviral infection. We found that in the commonly used C57BL/6 substrains of mice, where murine leukaemia virus (MLV) envelope glycoproteins are expressed by several endogenous MLV proviruses, the highest expressed MLV envelope glycoprotein is under the control of an immune-responsive cellular promoter, thus linking MLV envelope glycoprotein expression with immune activation. We further showed that antibody ligation induces extensive internalisation from the plasma membrane into endocytic compartments of MLV envelope glycoproteins, which are not normally subject to constitutive endocytosis. Importantly, antibody binding and internalisation of MLV envelope glycoproteins initiates signalling cascades in envelope-expressing murine lymphocytic cell lines, leading to cellular activation. Similar effects were observed by MLV envelope glycoprotein ligation by its cellular receptor mCAT-1, and by overexpression in human lymphocytic cells, where it required an intact tyrosine-based YXXΦ motif in the envelope glycoprotein cytoplasmic tail. Together, these results suggest that signalling potential is a general property of retroviral envelope glycoproteins and, therefore, a target for intervention.

## Author summary

The outcome of viral infection depends on the balance between host immunity and the ability of the virus to avoid, evade or subvert it. The envelope glycoproteins of diverse viruses, including retroviruses, are displayed on the surface of virions and of infected cells

gov/bioproject/PRJEB36787/) and ERP120023
(Jurkat cells; https://www.ncbi.nlm.nih.gov/
bioproject/PRJEB36792/).

**Funding:** This work was supported by the Francis
Crick Institute (FC001099 to GK and FC001162 to
JPS), which receives its core funding from Cancer
Research UK, the UK Medical Research Council
and the Wellcome Trust; and by the Wellcome
Trust (102898/B/13/Z to GK). The funders had no
role in study design, data collection and analysis,
decision to publish, or preparation of the
manuscript.

**Competing interests:** The authors have declared
that no competing interests exist.

and thus constitute the major target of the host antibody response. Antibody responses
are elicited not only against infectious viruses we acquire during our life-history, but also
against the numerous retroviral envelopes encoded by our genome and acquired during
our species' life-history. In turn, viruses have evolved ways to reduce exposure of their
envelope glycoproteins to the host immune system, including constitutive endocytosis or
antibody-induced internalisation. Using murine leukaemia viruses as models of infectious
and endogenous retroviruses, we show that antibody binding to retroviral envelopes
induces extensive internalisation of the envelope-antibody complex and initiates signal-
ling cascades, ultimately leading to transcriptional activation of envelope glycoprotein-
expressing lymphocytes. We further show that expression of endogenous retroviral enve-
lopes is coupled to physiological lymphocyte activation, integrating them with the
immune response. These findings reveal an unexpected layer of interaction between the
host antibody response and retroviral envelope glycoproteins, which could be considered
immune receptors.

## Introduction

The outcome of viral infection depends on the balance between host defences and virus' ability
to counteract, avoid or exploit them [1, 2]. Like all other viral infections, retroviral infection
triggers the production of antibodies against multiple retroviral proteins, but the principal tar-
get of protective antibodies is the retroviral envelope glycoproteins that comprise the envelope
spike (referred to here as envelope), presented on the surface of virions and infected cells [3–
5]. Indeed, envelope-specific antibodies are readily induced by human immunodeficiency
virus (HIV)-1/2 and human T-cell leukaemia virus (HTLV)-1/2 infection in humans, where
they are diagnostic of infection [6, 7], and by murine leukaemia virus (MLV) or mouse mam-
mary tumour virus (MMTV) infection in mice [5, 8].

  However, the envelopes of exogenous infectious retroviruses are not the only ones targeted
by the host antibody response. Both the human and mouse genomes contain several endoge-
nous retroviral *env* genes that have retained the potential to express full-length envelopes [9–
15]. Indeed, several envelopes of endogenous retroviruses (ERVs) are known to be expressed
in human and mouse cells under physiological conditions, as well as in pathologies such as
cancer, infection or autoimmunity, where expression can be upregulated [16, 17]. In addition
to the repurposed Syncytin genes, these include envelopes of human endogenous retrovirus
(HERV)-K, HERV-T and HERV-R in humans and of MLV, GLN and MMTV in mice [9–15].
Spontaneous induction of antibodies to human endogenous retroviral envelopes has been
amply documented in healthy humans and their levels may increase in systemic lupus erythe-
matosus (SLE) or cancer patients [13, 15, 18–24]. Similarly, antibodies to murine endogenous
retroviral envelopes can be spontaneously induced in healthy mice with age and have been
linked with disease severity in SLE mouse models [25, 26].

  Envelope-specific antibodies can neutralise viral infectivity by blocking the interaction with
the cellular receptor and also induce antibody-dependent cellular cytotoxicity (ADCC) and
complement-dependent cytotoxicity (CDC) [27–29]. However, retroviruses have evolved
diverse strategies to evade the action of envelope-specific antibodies, including a high muta-
tion rate and conformational or carbohydrate-shield masking of critical epitopes from neutral-
ising antibodies [30–32]. Certain retroviruses evade most actions of antibodies, simply by
reducing the amount of envelope accessible for antibody binding [33]. Effective antibody
responses against HIV-1 are thwarted by low expression of envelope both on the surface of

virions and of infected cells [34–36]. Surface envelope expression of HIV-1 and of other lentiviruses is thought to be the result of constitutive endocytosis from the plasma membrane of infected cells, a process that relies on a tyrosine-based motif (YXXΦ, where X represents any amino acid and Φ a bulky hydrophobic amino acid) in the envelope cytoplasmic domain, acting as an endocytosis signal [37, 38].

This motif is highly conserved among retroviruses independently of their host species or tropism [38] and, in addition to endocytosis, it can also direct envelope glycoproteins to specific regions of infected cell's plasma membrane, particularly in polarised cells [39]. The latter function of the YXXΦ sequence, acting as a sorting motif, concentrates viral budding at specific regions of the producer cell and is especially important for the virological synapse during cell-to-cell transmission of HTLV-1, another adaptation for evasion of the antibody response [40, 41].

Conservation of this motif extends to all four groups of MLV envelope, defined by their tropism and sequence as ecotropic, xenotropic, polytropic and modified polytropic [42], albeit it does not seem to promote endocytosis or affect cell surface levels of these envelopes [43]. It is therefore possible that conservation among retroviruses is due to its sorting motif functions, which would involve interaction with cytoskeletal or other membrane proteins. Despite the potential of retroviral envelopes to engage sorting and endocytosis pathways, the impact of their intracellular and plasma membrane trafficking patterns on host cell physiology is incompletely understood. Moreover, retroviral envelopes on infected cells are constantly engaged by antibodies that do not impact retroviral replication, either owing to viral evasion mechanisms, in the case of infectious exogenous retroviruses, or to the absence of replicating virus, in the case of human ERVs. Antibody ligation of retroviral envelopes or of envelope glycoproteins of other viruses can lead to internalisation of the envelope-antibody complexes [44–46]. However, the cell-intrinsic effects on host cell physiology of envelope ligation by such antibodies are unclear. Here, we used a reductionist approach to examine the causes and cell-autonomous consequences of MLV envelope expression. Our data suggest that overexpression of MLV envelope or ligation by antibodies or its cellular receptor can be a lymphocyte signalling initiating event.

## Results

### Full-length retroviral envelope expression in murine lymphocytes

The mouse genome contains several fully-codogenic retroviral *env* genes, with the potential to express in any cell type [47]. Analysis of public RNA sequencing (RNA-seq) data from resting and CD3/CD28 activated CD4[+] T cells from B6 mice [48], indicated constitutive transcription of several endogenous MLV *env* genes, primarily of xenotropic proviruses, as well as the single ecotropic provirus *Emv2* (Fig 1A). Of note, CD4[+] T cells expressed at the highest levels *Xmv45*, a xenotropic provirus previously found the most highly expressed also in activated B cells [49]. Inspection and junctional analysis of the *Xmv45 env* transcript in resting and LPS-activated B cells revealed that it was not driven by the proviral LTR, but by the promoter of the lncRNA *AI506816*, which spliced directly to the *Xmv45 env* gene, and similar results were also obtained with CD4[+] T cells (Fig 1B). It was therefore likely that the responsiveness of *Xmv45 env* to immune stimulation was provided by the *AI506816* promoter. To examine the responsiveness to immune stimuli, we used LPS stimulation of B cells, since *Xmv45* transcription was not altered by CD3/CD28 stimulation of T cells (Fig 1A). As *Xmv45* is insertionally polymorphic between mouse strains, we examined the LPS-responsiveness of the ancestral *AI506816* gene (provisionally referred to as *AI506816a*), prior to *Xmv45* integration, using *Xmv45*- or *AI506816a*-specific primers (S1A Fig). As expected [49, 50], *Xmv45 env* was induced by LPS

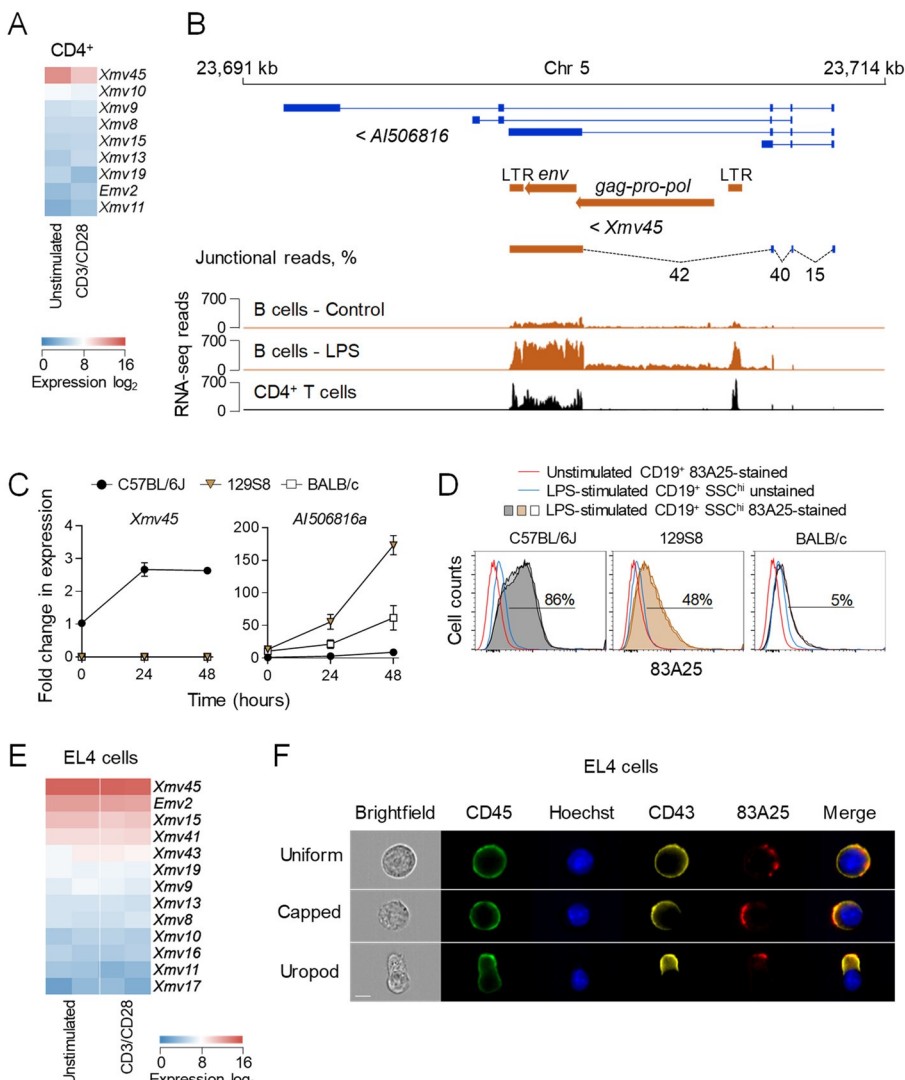

**Fig 1. An endogenous MLV *env* is an immune stimulation-responsive gene in B cells and is expressed by primary lymphocytes and EL4 thymoma cells.** (A) Heatmap of endogenous MLV expression in resting and CD3/CD28-stimulated CD4+ T cells assessed by RNA-seq. (B) Genomic location of *Xmv45* and RNA-seq read mapping in unstimulated and LPS-stimulated B cells and in unstimulated CD4+ T cells. (C) Splenocytes of C57BL/6J, 129S8 or BALB/c mice were *in vitro* stimulated with LPS for 24 or 48 hours and the expression of *Xmv45* (left) and *AI506816a (right)* was assessed by qRT-PCR. Expression levels following LPS stimulation were compared to basal expression levels in unstimulated C57BL/6J splenocytes. (D) Flow cytometric analysis of MLV envelope on the surface of SSChi B cells from C57BL/6J, 129S8 or BALB/c mice following *in vitro* stimulation with LPS for 48 hours compared with unstimulated, but stained B cells and with stimulated, but 83A25-unstained B cells. Two mice per strain are shown. (E) Heatmap of endogenous MLV expression in resting and CD3/CD28-stimulated EL4 cells assessed by RNA-seq. Two biological replicates are shown. (F) MLV envelope localisation in non-polarised and polarised EL4 cells. EL4 cells were labelled with anti-CD45, anti-CD43 for 30 minutes and 83A25 for 15 minutes at 37˚C, counterstained with Hoechst and imaged by IS. Scale bar = 7 μm.

stimulation of B6 B cells, but not those from 129S8 or BALB/c mice, which lack the *Xmv45* integration (Fig 1C and S1B Fig). In contrast, *AI506816a* was strongly induced by LPS in 129S8 and BALB/c B cells (Fig 1C). Accordingly, LPS-inducible expression of MLV envelopes, detected by the 83A25 antibody, which detects a common epitope in all endogenous MLV envelopes [51], was reduced or absent in 129S8 and BALB/c B cells (Fig 1D). These results

suggested that the chimeric *AI506816-Xmv45* transcript was a major source of the MLV envelope expression in primary B6 lymphocytes. *Xmv45* was also the highest expressed endogenous MLV in the EL4 murine T cell lymphoma cell line (Fig 1E), which is free from infectious MLVs [52]. Similarly to primary CD4$^+$ T cells, *Xmv45* expression in EL4 cells was constitutive and was not further increased by CD3/CD28 stimulation (Fig 1E). MLV envelope demarcated distinct plasma membrane sites in EL4 cells with no apparent polarity and accumulated in the uropod of polarised EL4 cells, co-localising with CD43 (Fig 1F), a marker for the uropod [53–55]. Non-uniform envelope expression was observed also in the plasma membranes of 2695 murine B cell lymphoma cells expressing infectious eMLV [52], and Jurkat human T cell lymphoma cells transduced to express *Emv2 env* (Jurkat.Emv2env) (S2 Fig), indicating it was a general property of MLV envelopes.

## Antibody-induced MLV envelope glycoprotein internalisation

In contrast to lentiviral envelopes, surface levels of which are regulated by endocytosis, MLV envelopes are not subject to constitutive endocytosis [43]. Nevertheless, it was conceivable that MLV envelope localisation is regulated by specific antibodies that are often induced against endogenous and exogenous retroviruses. Indeed, ligation of MLV envelopes with the 83A25 antibody led to internalisation of the complex in EL4 cells, reaching punctate intracellular compartments over several hours (Fig 2A). Identical findings were obtained with a number of EL4 sublines and B cell and pre-B cell leukaemia cell lines that carry infectious viruses (S3 Fig), and with uninfected primary naïve and CD3/CD28-simulated T cells and primary naïve and LPS-stimulated B cells (Fig 2B). Of note, internalisation of envelope-antibody complexes in activated T and B cells also correlated with internalisation of CD3 and CD19, signalling receptors in T and B cells, respectively, which co-localised with internalised envelope (Fig 2B). Envelope-antibody internalisation was also observed with E.G7-OVA cells incubated with the xMLV envelope-specific antibody 522 [56] and with EL4 cells expressing Friend-MLV (F-MLV) clone FB29 envelope (EL4.FB29env) incubated with the 720 antibody, which is specific to F-MLV envelope and does not cross-react with endogenous MLV envelopes [57] (S4A and S4B Fig). It was also observed with Jurkat.Emv2env cells incubated with the 83A25 antibody (S4C Fig). In contrast, antibody-ligated mouse or human CD45 was not internalised in EL4 or Jurkat.Emv2env cells, respectively, during the same time-frame (S5 Fig). Antibody-induced envelope internalisation caused loss of plasma membrane envelope, as staining of E. G7-OVA cells with the non-competing 522 antibody was substantially reduced following internalisation with the 83A25 antibody (Fig 2C) and directed the complexes to acidified intracellular compartments (Fig 2D), visualised by conjugating the 83A25 antibody with pHrodo, which fluoresces only in acidic pH. These results demonstrated specific envelope-antibody complex internalisation in a variety of cell, antibody and MLV envelope combinations.

## Envelope glycoprotein antibody binding initiates signalling

Given the potential interaction of retroviral envelopes with range of host membrane-proximal proteins, it was possible that antibody binding and internalisation of envelope affected proximal or associated proteins and, by extension, cellular processes. Inspection of 83A25-incubated EL4 cell cultures revealed strong clustering, reminiscent of T cell activation, which was not observed upon incubation with an isotype control or an anti-CD5 antibody (Fig 3A). Clustering was also observed with A1 pre-B cell leukaemia cells following incubation with the 83A25 antibody (S6 Fig). 83A25-induced EL4 cell clustering was dependent on LFA-1 (Fig 3B), consistent with the established role for integrins in T cell adhesion [58]. Supporting this observation, EL4 cells incubated with the 83A25 antibody, but not with an anti-CD5 antibody,

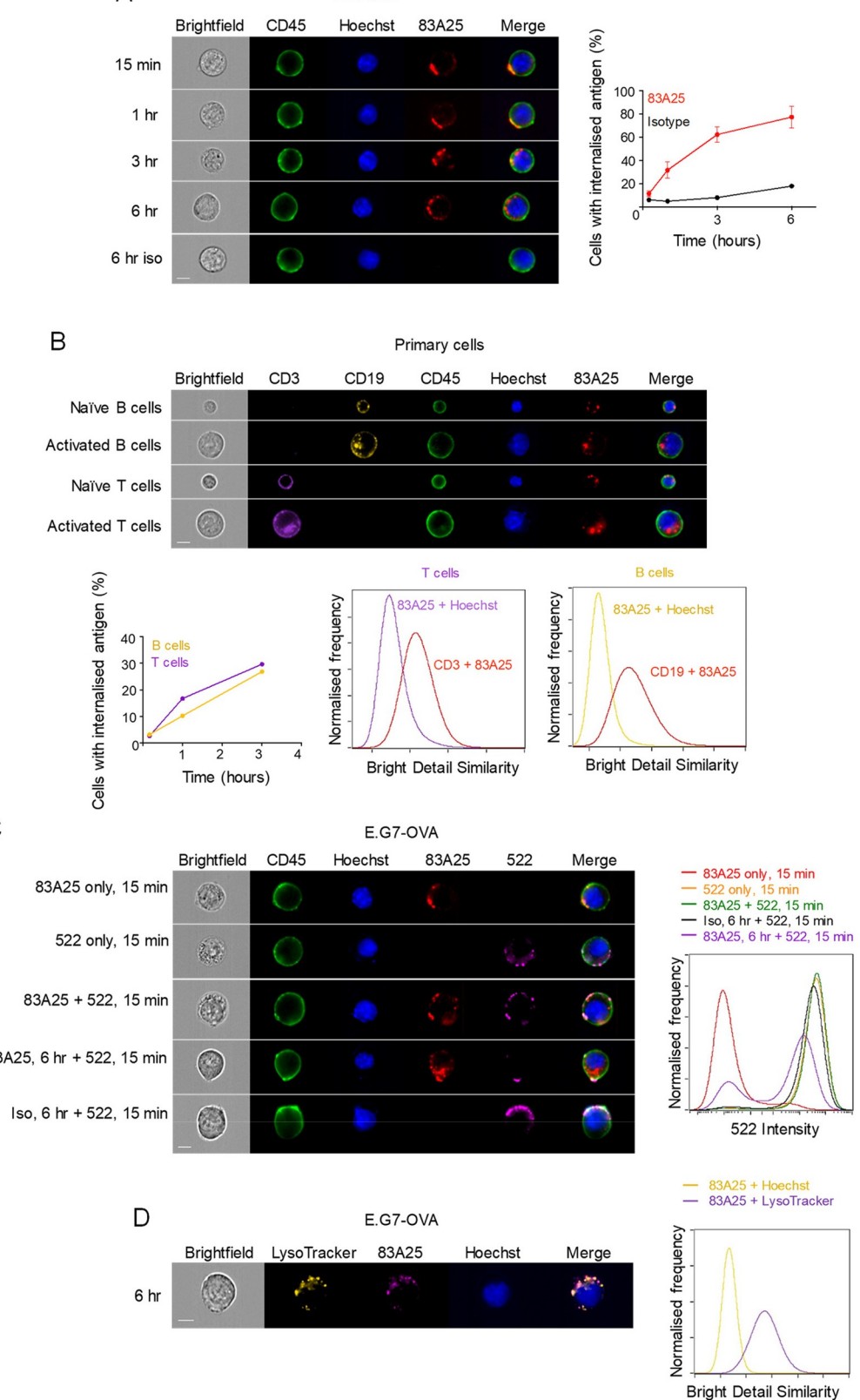

**Fig 2. Endogenous MLV envelope is internalised by EL4 cells and primary lymphocytes.** (A) IS images of EL4 cells incubated with 83A25 antibody for specified periods of time and counterstained with anti-CD45 and Hoechst (left). Percentage of cells with internalised envelope-antibody complexes (right) over time from four independent experiments. A minimum of 5000 cells were analysed in each experiment at each time point. Scale bar = 7 μm. (B) Splenocytes were imaged by IS in the naïve state or were activated with LPS or CD3/CD28 Dynabeads for 48 hours. Prior to imaging, cells were incubated with 83A25 antibody for specified periods of time. At the end of incubation with 83A25, cells were labelled with anti-CD45, anti-CD3, anti-CD19 and Hoechst. Three-hour time point images are shown (top). Percentage of activated cells with internalised env-antibody complexes at specified time points (bottom left). Co-localisation of 83A25 with CD19 in B cells and CD3 in T cells (bottom right) was quantified using the Bright Detail Similarity feature in IDEAS and compared to Hoechst, a non-colocalising probe. A minimum of 10000 cells were analysed at each time point. Scale bar = 7 μm. (C) Antibody binding to MLV envelope reduces the number of envelope molecules present on the cell surface. E.G7-OVA cells were incubated with either 83A25, 522, isotype control antibody or a combination of these for defined periods of time and counterstained with anti-CD45 and Hoechst. Images (left) and fluorophore signal intensities (right) were recorded by IS. Data representative of three independent experiments with a minimum of 10000 cells analysed per treatment group. Scale bar = 7 μm. (D) Internalised envelope-antibody complexes localise to acidic endosomal/lysosomal compartments. IS images of E.G7-OVA cells incubated with pHrodo-conjugated 83A25 for three hours and stained with LysoTracker and Hoechst dyes (left). Co-localisation of 83A25 with LysoTracker was quantified using the Bright Detail Similarity feature in IDEAS and compared to Hoechst, a non-colocalising probe (right). Scale bar = 7 μm.

exhibited evidence of degranulation, measured by the cell surface exposure of CD107a (Fig 3C). EL4 cell degranulation was also observed when eMLV envelope on EL4 cells was ligated by its cellular receptor mCAT-1 by co-incubation with 293T cells expressing mCAT-1 (293T. mCAT-1), but not with parental 293T cells (Fig 3D).

To further probe potential activation of EL4 cells by envelope ligation, we analysed their transcriptional response following 83A25 stimulation. Parallel stimulation with CD3/CD28 and even isotype control antibodies had a noticeable effect on EL4 cell transcriptional state (S7 Fig). Nevertheless, stimulation with 83A25 modulated a distinct set of genes, not shared with the isotype control and only partially shared with CD3/CD28 (Fig 3E). These included *Nhsl2*, *Tgfb3*, *Abhd12b*, *Pmaip1* and other genes, specific induction of which was validated by qRT-PCR (Fig 3F and S8 Fig). We focused on transcription of *Nhsl2* and *Tgfb3*, which were two of the most responsive genes specifically to 83A25 stimulation to address a number of questions. Assessed by transcription of these two genes, the response of EL4 cells to 83A25 stimulation was neither blocked by the FcR-binding antibody 2.4G2, nor observed with the 2.4G2 antibody alone (S9A Fig), suggesting that this property of the 83A25 antibody did not depend on FcR binding. FcR dependency could not be assessed with 83A25 antigen-binding fragments (Fab), as they failed to bind MLV envelope on EL4 cells with sufficient affinity. Induction of *Nhsl2* and *Tgfb3* transcription was also observed when EL4.FB29env cells were stimulated with the 720 antibody, but not when EL4 cells were stimulated with the 522 antibody (S9B Fig). These findings indicated that signalling required either engagement of an eMLV envelope, such as *Emv2* and FB29 (recognised by 83A25 and 720, respectively) or IgG antibody class (such as 83A25 or 720) and was not efficiently induced by IgM antibodies (such as the xMLV-specific 522). *Nhsl2* and *Tgfb3* transcriptional induction was also observed when EL4 cells were cultured with 293T.mCAT-1, but not parental 293T cells (S9C Fig), where eMLV envelope on EL4 cells would engage with its cognate receptor on target 293T.mCAT-1 cells. Internalisation of envelope-antibody complexes seemed necessary to induce *Nhsl2* and *Tgfb3* transcription in EL4 cells, as immobilisation of the 83A25 antibody on plastic or beads was ineffective (Fig 4A and 4B). This was not due to reduced availability of the immobilised 83A25 antibody, as its soluble form efficiently induced *Nhsl2* and *Tgfb3* transcription even at over 10 times lower concentration (S10 Fig), and immobilised anti-CD3 efficiently induced *Il2* transcription under the same conditions (Fig 4A and 4B). Conversely, incubation of EL4 cells with an antibody against CD5, which is known to be internalised [59], led to efficient internalisation of the complex, but did not induce *Nhsl2* and *Tgfb3* transcription (S11 Fig). These

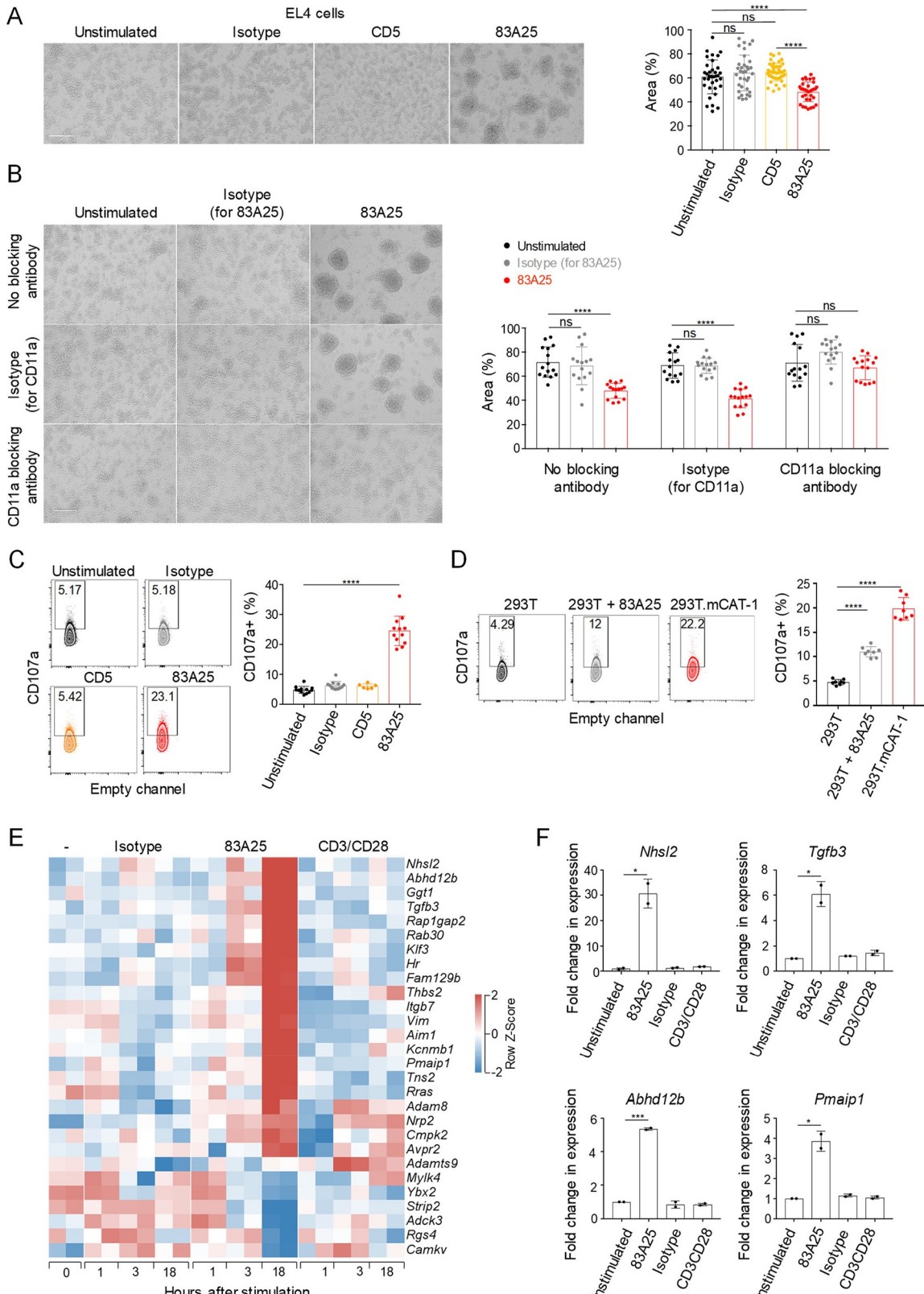

**Fig 3. Stimulation of EL4 cells with 83A25 induces cell clustering and transcriptional changes.** (A) EL4 cells cluster following incubation with 83A25. Light microscopy images of EL4 cells incubated with indicated antibodies for 18 hours (left). Scale bar = 200 μm.

Quantification of area occupied by the cells as a percentage of total per field of view (right). Pooled data from three independent experiments with at least ten fields of view per experiment. (B) Blocking LFA-1 α subunit (CD11a) prevents EL4 cluster formation. Light microscopy images of EL4 cells incubated with indicated antibodies for 18 hours (left). Scale bar = 200 μm. Quantification of area occupied by the cells as a percentage of total per field of view (right). Pooled data from three independent experiments with at least five fields of view per experiment. (C) EL4 cells were incubated with indicated antibodies for 18 hours and assessed for surface CD107a by flow cytometry. Representative flow cytometric plots (left) and quantitation of CD107a positive cells as a percentage of total (right). Pooled data from three independent experiments. (D) EL4 cells were co-cultured with 293T or 293T.mCAT-1 cells in the absence or presence of 83A25 antibody for 18 hours and assessed for surface CD107a by flow cytometry. Representative flow cytometric plots (left) and quantitation of CD107a positive EL4 cells as a percentage of total (right). Pooled data from two independent experiments. (E) Heatmap of differentially expressed genes assessed by RNA-seq showing a comparison of untreated, isotype control antibody treated, 83A25 antibody treated or CD3/CD28 Dynabeads treated EL4 cells following 1, 3 or 18 hours of stimulation. Two biological replicates for each treatment and time point are shown. (F) Expression of *Nhsl2*, *Tgfb3*, *Abhd12b* and *Pmaip1* genes assessed by qRT-PCR in EL4 cells stimulated with 83A25 antibody for 18 hours.

results indicated that internalisation of envelope-antibody complexes was necessary, whereas internalisation of CD5-antibody complexes was not sufficient to induce *Nhsl2* and *Tgfb3* transcription.

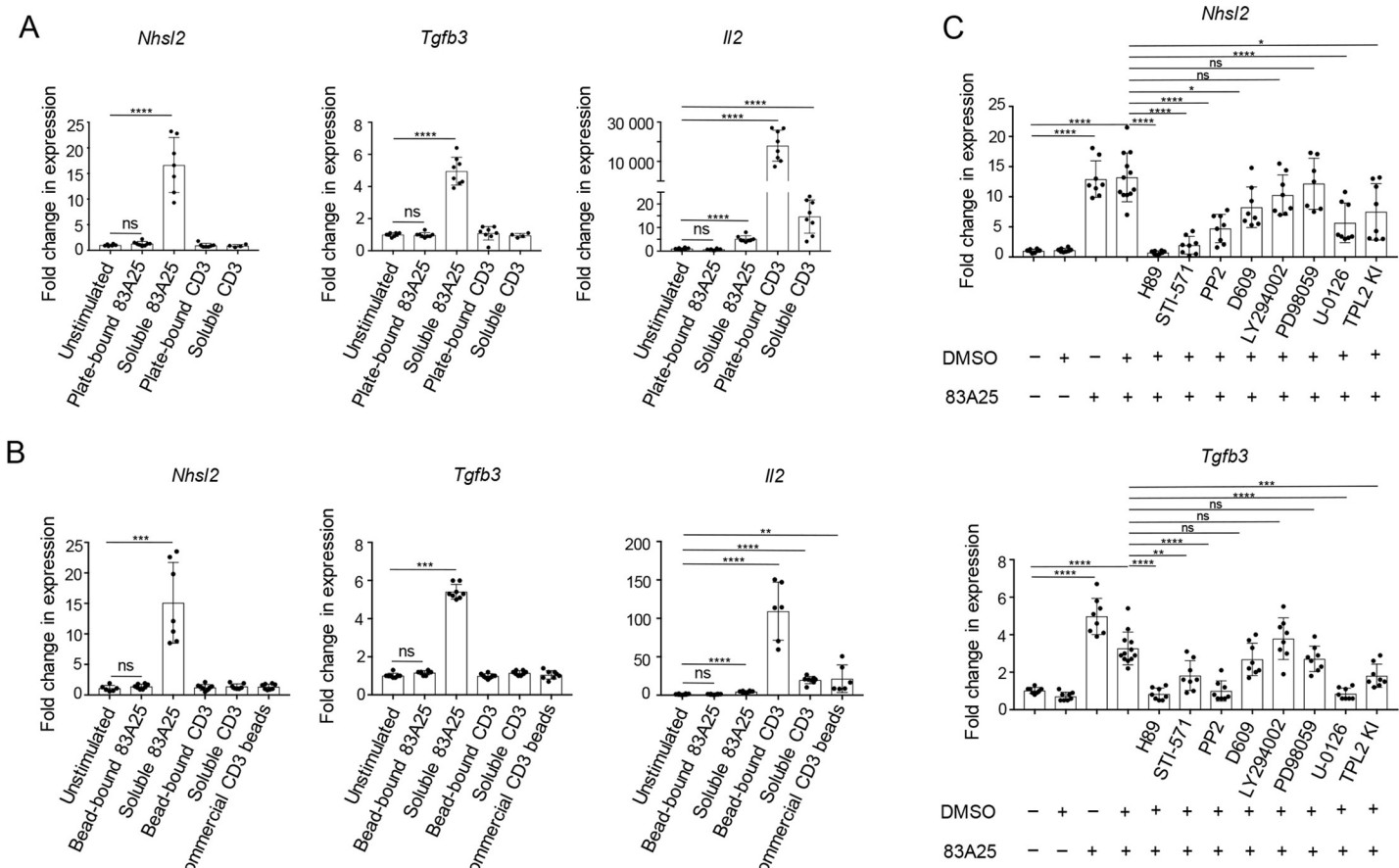

**Fig 4. Internalisation of MLV envelope-antibody complexes is necessary for the initiation of intracellular signalling.** (A) Expression of *Nhsl2*, *Tgfb3* and *Il2* genes assessed by qRT-PCR in EL4 cells stimulated with plate-bound or soluble 83A25 antibody for 18 hours. Pooled data from two independent experiments. (B) Expression of *Nhsl2*, *Tgfb3* and *Il2* genes assessed by qRT-PCR in EL4 cells stimulated with bead-bound or soluble 83A25 antibody for 18 hours. Pooled data from two independent experiments. (C) Chemical inhibition of signalling in EL4 cells. EL4 cells were either left untreated or pre-treated with indicated inhibitors for 5 minutes prior to addition of the stimulating 83A25 antibody. Cells were incubated for 18 hours and assessed for *Nhsl2* and *Tgfb3* expression by qRT-PCR. Pooled data from two independent experiments.

These phenotypic and transcriptional changes induced by 83A25 stimulation of EL4 cells were consistent with the utilisation of signalling cascades following envelope ligation and internalisation. To investigate putative signalling cascades mediating the transcriptional response of EL4 cells to 83A25, we used a range of signalling inhibitors (Fig 4C). The protein kinase A (PKA) inhibitor H89 was the most effective in preventing *Nhsl2* and *Tgfb3* transcriptional induction, followed by the STI-571 (Abl kinase) and PP2 (Src kinase) tyrosine kinase inhibitors (Fig 4C). In contrast, inhibitors of phospholipase C-γ (PLC-γ) and phosphoinositide 3-kinases (PI3K), D609 and LY294002, respectively, were ineffective (Fig 4C). The MEK1 kinase inhibitor PD98059 also had limited effects, but U-0126, an inhibitor of both MEK1 and MEK2 and a TPL2 kinase inhibitor had a significant, albeit partial effect (Fig 4C). Consistent with the dependence of 83A25 stimulation on functioning PKA, Abl and Src signalling pathways, EL4 cells exhibited high levels of phosphorylated targets of these pathways, such as CREB and ERK (S12 Fig). However, CREB and ERK activation was constitutively present in EL4 cells and was not increased further by 83A25 or CD3/CD28 stimulation (S12 Fig). Collectively, these results argue that, despite their constitutive activation leading to CREB and ERK phosphorylation, signalling by PKA and tyrosine kinases, but not PLC/PI3K/Akt, is required to couple these pathways to transcriptional activation of *Nhsl2* and *Tgfb3* in response to envelope ligation.

## Transcriptional activation of human cells by eMLV envelope over-expression

To extend these observations in a separate system, we examined the signalling capacity of eMLV envelope expressed in Jurkat.Emv2env cells. The choice of human cells precludes confounding effects of multiple endogenous MLV envelopes expressed in murine cells or of other envelope-interacting MLV proteins, such as Gag, and allows the study of precise envelope mutants. As human cells express receptors for xMLV, but not eMLV envelopes, we chose to express the latter.

Surprisingly, over-expression of eMLV envelope in Jurkat.Emv2env cells led to substantial alteration of their transcriptional program, compared with parental Jurkat cells, with over 1,200 genes differentially expressed between the two types of cell (S1 Table). These genes were involved in processes related predominantly to plasma membrane organisation (S13 Fig), suggesting that expression of this transmembrane envelope glycoprotein, modified its environment also through altered gene expression. Several of the responsive genes were known to be modulated during T cell development or activation (Fig 5A). For example, the upregulation of *SELL* (encoding L-selectin) and *IL7R* and downregulation of genes involved in TCR rearrangement and selection, such as *RAG1*, *RAG2* and *PTCRA* (encoding the pre-TCRα chain), typifies T cell development, and *EGR1* and *DLX3* are induced during T cell activation, also in Jurkat cells [60]. Indeed, RNA-seq analysis over the course of stimulation revealed upregulation of *EGR1* and *DLX3* and downregulation of *RAG1*, *RAG2* and *PTCRA* in Jurkat.Emv2env cells stimulated with 83A25 or CD3/CD28, but not in unstimulated cells or those stimulated with an isotype control antibody (Fig 5B). These results suggested transcriptional activation of Jurkat.Emv2env cells by the over-expression of eMLV envelope and further activation by its ligation with the 83A25 antibody. This transcriptional signature of eMLV envelope over-expression, exemplified in the transcription of *FLT3*, *NELL1* and *PRKG2*, was proportional to the level of eMLV envelope expression and was not present in Jurkat T cells transduced to express only GFP (Jurkat.GFP cells) (S14A and S14B Fig). Moreover, this signature was induced not only in established Jurkat.Emv2env sublines, but also in unselected Jurkat cells

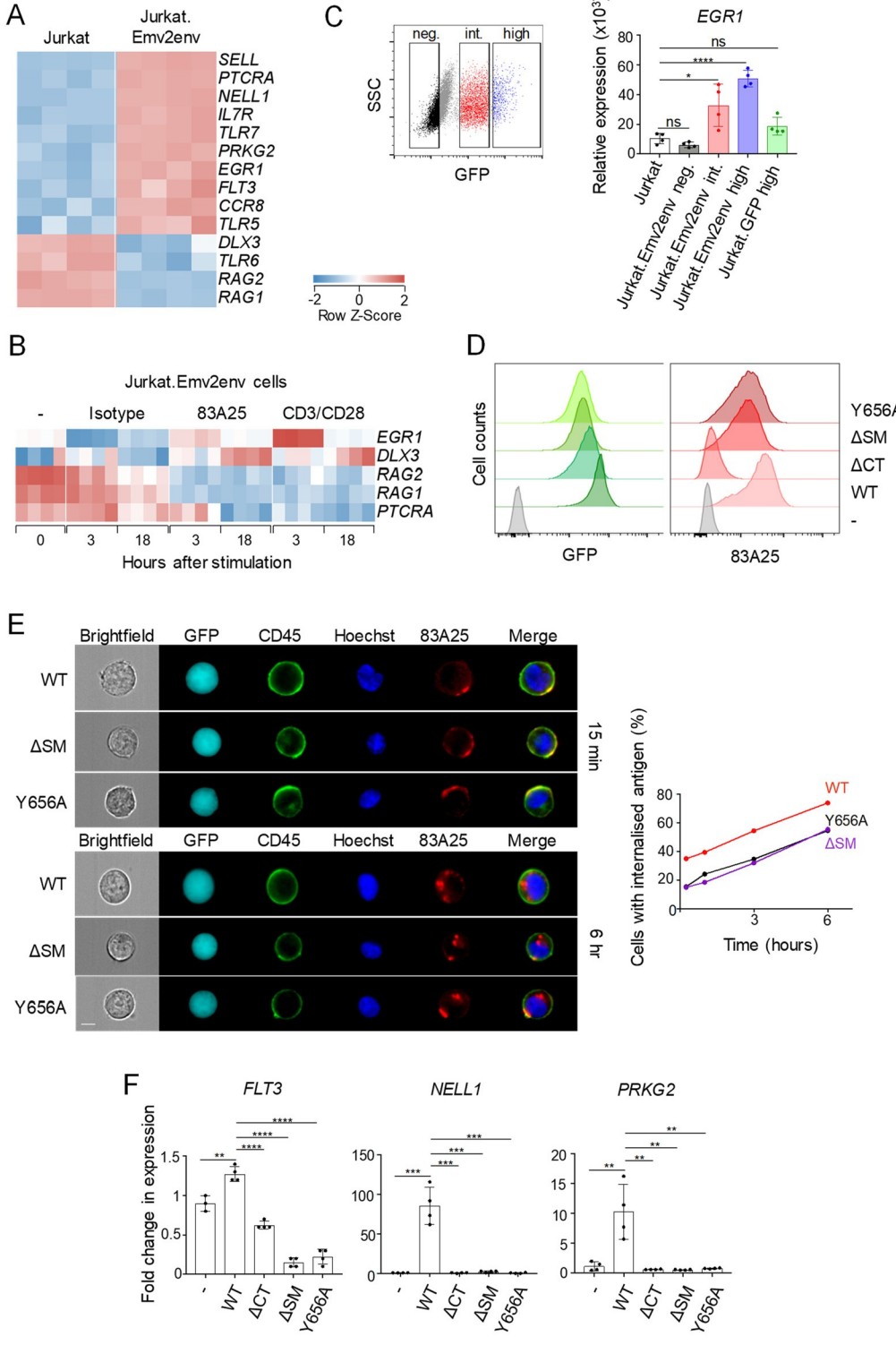

**Fig 5. Sorting motif within the cytoplasmic tail of MLV envelope is required for intracellular signalling.** (A) Heatmap of genes differentially expressed by Jurkat and Jurkat.Emv2env cells assessed by RNA-seq. Four biological replicates are shown. (B) Heatmap of genes affected by 83A25 antibody treatment in Jurkat.Emv2 cells assessed by RNA-seq at 3 hours and 18 hours post-stimulation. Four biological replicates are shown. (C) Jurkat cells were transduced with a retrovirus carrying the *Emv2 env* gene and a gene encoding GFP as a selectable marker, in a bicistronic genome. Three to four days following transduction cells were sorted as GFP negative, GFP intermediate or GFP high (left). Cells transduced with a retrovirus carrying only the GFP encoding gene were sorted for high

expression of GFP and were also included. Sorted populations were assessed for *EGR1* expression by qRT-PCR (right). (D) Flow cytometric analysis of GFP and Emv2 envelope expression in Jurkat.Emv2env WT, Jurkat.Emv2env ΔCT, Jurkat.Emv2env ΔSM and Jurkat.Emv2env Y656A cells. (E) IS images of Jurkat.Emv2env WT and mutant cells incubated with 83A25 antibody for 15 minutes and counterstained with anti-CD45 and Hoechst showing envelope distribution on the cell surface (top panel, left). IS images of Jurkat.Emv2env WT and mutant cells incubated with 83A25 antibody for 6 hours and counterstained with anti-CD45 and Hoechst showing levels of envelope internalisation (bottom panel, left). Percentage of cells with internalised envelope-antibody complexes in Jurkat.Emv2env WT and mutant cells (right). A minimum of 10000 cells were analysed at each time point. Scale bar = 7 μm. (F) Expression of *FLT3*, *NELL1* and *PRKG2* genes assessed by qRT-PCR in Jurkat.Emv2env WT and mutant cells.

shortly after transduction with the *Emv2 env*-expressing vector, again according to eMLV envelope expression levels, but not with a GFP-expressing vector (Fig 5C).

## Intact sorting motif required for eMLV envelope-mediated transcriptional activation

The findings that eMLV envelope over-expression in Jurkat T cells modified their basal transcriptional state offered the opportunity to study the requirements for potential envelope signalling or interacting domains without any confounding effects of antibody binding and complex internalisation. We therefore over-expressed in these cells either the WT *Emv2* envelope or variants with alanine substitution of the tyrosine in the sorting motif (Y656A), deletion of the sorting motif (ΔSM) or deletion of the entire cytoplasmic domain (ΔCT).

Jurkat sublines were established expressing these constructs at comparable levels, assessed by expression of an internal ribosomal entry site (IRES)-GFP reporter gene fused to the *env* cDNA (Fig 5D). Mutation or deletion of the SM did not overly affect eMLV envelope expression at the plasma membrane (Fig 5D), consistent with prior reports [43]. In contrast, deletion of the cytoplasmic domain prevented surface expression of eMLV envelope, which was retained intracellularly (Fig 5D; S15 Fig). Surface distribution of ΔSM and Y656A mutants of eMLV envelope was more localised than the WT equivalent and internalisation following incubation with the 83A25 antibody was reduced, but not prevented (Fig 5E). Importantly, however, expression of neither of the three eMLV envelope mutants induced the transcription of indicator genes *FLT3*, *NELL1* and *PRKG2*, which were induced by expression of the WT counterpart (Fig 5F), suggesting that the residual internalisation of the mutants was not sufficient for signalling and highlighting a critical requirement for an intact sorting motif. Together, these results further supported the notion that localisation at the plasma membrane and interaction of MLV envelope glycoproteins with host cell proteins was necessary to initiate a signalling cascade, dependent on the function of the sorting motif.

## Discussion

Despite the frequent expression of endogenous retroviral envelopes in diverse cell types of the mammalian host and the induction of envelope-binding antibodies [13, 17–20], the consequence of their interaction for cell physiology are still incompletely understood. Here, we provide evidence to suggest that ligation of MLV envelopes by antibodies or the cellular receptor initiates a signalling cascade leading to transcriptional and phenotypic activation of lymphocytic cell lines.

Given their potential interaction with a number of cytoplasmic and transmembrane cellular proteins, it is possible that retroviral envelopes initiate or alter signalling cascades and alter cell physiology. One clear example is the envelope of Jaagsiekte sheep retrovirus (JSRV), which acts as an oncogene, leading to the activation of PI3K/Akt and MAPK signalling cascades and directly causing lung adenocarcinomas in sheep [61]. More recently, envelopes encoded by the

HML2 family of HERV-K proviruses have been shown to activate the ERK1/2 signalling pathway when overexpressed in HEK293T cells, a phenotype dependent on an intact envelope cytoplasmic tail, and to contribute to the oncogenic properties of human breast epithelial cell lines [62]. However, this property of HERV-K(HML2) envelopes was not shared by envelopes of other retroviruses examined in that study, including MLV [62]. Ectopic expression of Syncytin, the envelope encoded by a HERV-W *env* gene, in astrocytes has also been shown to induce production of proinflammatory mediators, although a precise signalling cascade has not yet been elucidated [63]. It may, therefore, be the case that signalling capacity is a general feature of retroviral envelopes, with distinct effects, both in degree and type, in different cell types. In support of this notion, we found that the overexpression of the murine *Emv2* envelope in human lymphocytic cells induced an extensive transcriptional response, in proportion with envelope expression levels and entirely dependent on the tyrosine-based YXXΦ motif in the envelope cytoplasmic tail. It is conceivable that interaction of this motif with different adaptor proteins in different cells engages retroviral envelopes in distinct cellular pathways and signalling cascades.

The YXXΦ motif may also be responsible in the regulation of envelope levels on the cell surface, at least for certain retroviruses [37, 38]. Low overall density of envelope at the plasma membrane appears to be a general feature of retroviral infection, contributing to evasion of host immunity [34–36]. However, its mechanism of action differs between retroviruses. In one group, including HIV-1 and HTLV-1, the envelope concentration is kept low through YXXΦ-dependent endocytosis after it reaches the plasma membrane [37, 39, 40, 43]. In contrast, in another group, including MLVs, the envelope is retained in the Golgi and mutations in the YXXΦ motif do not appreciably alter its total amount at the plasma membrane [43], consistent with the results of this study.

Although viral replication necessitates envelope expression at the plasma membrane, successful virion production and cell-to-cell infection are facilitated by localised expression of envelope at distinct plasma membrane domains [39–41, 64]. The tyrosine-based YXXΦ motif seems to be centrally involved in this process too. Whilst not controlling endocytosis of MLV envelopes, the YXXΦ motif controls basolateral release of infectious virions in polarised cells in diverse retrovirus groups, including MLV [38], suggesting interaction with intracellular and membrane-proximal host cell proteins. This effect was dependent on the tyrosine in the YXXΦ motif of MLV envelope, but not its phosphorylation [65, 66], suggesting an adaptor function for this motif.

The role of the YXXΦ motif in MLV replication in vivo is unclear, as mutations of this motif did not affect replication of the ecotropic Cas-Br-E strain of MLV in neonatally-infected mice [38]. However, such mutations did affect pathogenesis, with a delay in leukaemia onset and, more pertinently, involvement of lymph node and thymus in addition to the spleen in mice infected with YXXΦ motif-mutant MLVs, in contrast to exclusive splenomegaly in mice infected with WT MLV [38]. These findings suggest that the YXXΦ motif normally prevents induction of T cell leukaemias/lymphomas, observed only with YXXΦ motif-mutant MLVs [38]. Other strains of MLV bearing an intact YXXΦ motif, most notably Moloney MLV, induce predominantly T cell leukaemias/lymphomas, but the type of resulting tumour in these cases is controlled by U3 enhancer sequences in the viral long terminal repeat (LTR), rather than the envelope [67]. It would be interesting to explore a possible mechanistic link between envelope-induced signalling in T cell leukaemia/lymphoma cell lines here and the reported induction of T cell leukaemias/lymphomas with YXXΦ motif-mutant MLVs.

In addition to initiating signalling cascades when expressed at high levels, our findings suggest that MLV envelopes induce stronger transcriptional responses following ligation by antibodies or their cellular receptor. Such ligation leads to extensive internalisation of envelope-

antibody complexes, a process that appears necessary for signal initiation. It is interesting to note that apparent signalling following eMLV envelope ligation by its cognate receptor mCAT-1 on target cells might also be accompanied by envelope-mCAT-1 internalisation. Indeed, it was recently demonstrated that upon interaction with mCAT-1-expressing cells, eMLV envelope-expressing cells internalise fragments of the target mCAT1-expressing cells [68].

Extensive internalisation of MLV envelopes upon antibody ligation renders the cells invisible to anti-envelope antibodies or other arms of immunity dependent on antibodies, such as ADCC, and may therefore represent an immune-escape mechanism tuned to the levels of envelope-binding antibodies. Internalisation of antibody complexes with viral proteins expressed on infected cells extends beyond retroviruses and has been described for human Respiratory Syncytial Virus (RSV) fusion protein [45] and Feline coronavirus (FCoV) spike and membrane proteins [46]. Moreover, although plasma membrane levels of HIV-1 envelope are thought to be kept low through constitutive endocytosis, recent studies demonstrated antibody-induced internalisation also of the HIV-1 envelope [44]. Interestingly, HIV-1 envelope internalisation was found to be conformation-specific and induced primarily by antibodies recognising the 'closed' conformation of envelope, prior to CD4 binding [44]. These findings revealed that antibody binding to the two different HIV-1 envelope conformation have different fates, in turn suggesting conformation-depended interaction with the endocytosis machinery and transmission of a potential signal.

Internalisation of envelope-antibody complexes may additionally affect signalling by other lymphocyte receptors, collaterally internalised. For example, internalisation of MLV envelope after antibody ligation in primary mouse lymphocytes overlaps with internalisation of CD3 in T cells and CD19 in B cells, receptors transmitting activating signals in the respective lymphocyte type, but not of the signalling-inhibiting receptor CD45.

Human retroviral infection with lymphocyte-tropic HIV-1 or HTLV-1 is associated with the development of autoimmunity [69, 70]. Elevated expression of endogenous retroviruses has also been linked with autoimmune manifestations, particularly SLE, in humans and mice [13, 15, 18–26, 71]. Experimental graft-versus-host reaction in mice has long been known to elicit autoimmune antibodies, targeting primarily the envelopes of endogenous retroviruses [72, 73], further supporting the immunogenicity of these self-antigens of retroviral origin [74]. Antibodies reactive with endogenous xMLV envelopes are known to be pathogenic and induce autoimmune pathology upon transfer into non-autoimmune mice [75], and endogenous xMLV envelopes are strongly increased in the sera of autoimmune-prone mice as part of an acute inflammatory response to LPS injection {Shigemoto, 1992 #97}. Acute phase endogenous MLV envelope production is genetically determined, through the combination of the endogenous MLV proviral complement and the alleles at the *Gv1*/*Sgp3* locus that controls expression of some of these proviruses [25, 76, 77]. Although *Xmv45* belongs to the group of endogenous MLVs that are not subject to *Gv1*/*Sgp3* control [77] and is, therefore, unlikely to be a source of serum envelope, its responsiveness to immune stimulation bears remarkable similarities to other xMLV proviruses that have become integrated into the acute phase response, highlighting a more general theme. It is reasonable to speculate that non-neutralising antibodies that are almost invariably induced against exogenous and endogenous retroviral envelopes contribute to lymphocyte dysfunction or aberrant activation that characterises autoimmunity, which warrants further investigation. Moreover, endogenous retroviral envelopes are likely far more dispensable than other self-antigens and their downregulation, through epigenetic repression of the source proviruses, may offer a well-tolerated means of preventing their potential contribution to autoimmunity.

## Materials and methods

### Ethics statement

C57BL/6J, C57BL/6NCrl, BALB/c and129S8 mice were maintained at the Francis Crick Institute's animal facilities under specific pathogen-free conditions. All animal experiments were approved by the ethical committee of the Francis Crick Institute, and conducted according to local guidelines and UK Home Office regulations under the Animals Scientific Procedures Act 1986 (ASPA).

### Cell culture conditions, stimulation and inhibition assays

Cells were cultured in IMDM media (Sigma) supplemented with 5% FCS, 50 μM β-mercaptoethanol (Sigma), 2 mM L-glutamine (Sigma), and 10 units/mL penicillin/streptomycin (Thermo Fisher) at 37°C in a 5% $CO_2$ incubator. When indicated, antibodies were added to culture media at 10 μg/mL. B cell activation was achieved by stimulation with LPS (10 μg/mL, Enzo), T cells and T cell tumour lines were activated using mouse or human T-Activator CD3/CD28 Dynabeads (ThermoFisher) at 1:1 bead-to-cell ratio. Inhibition assays were set up in complete media and cells were incubated with the appropriate inhibitor for 5 minutes prior to addition of a stimulating antibody. All inhibitors were dissolved in DMSO as a 1000-fold stock, except PP2 which was dissolved as a 200-fold stock. Final inhibitor concentrations were 10 μM H89, 10 μM STI-571, 20 μM PP2, 20 μM D609, 5 μM LY294002, 30 μM PD98059, 10 μM U-0126, 5 μM Tpl2. All inhibitors were from Cambridge Bioscience.

### Antibody immobilisation assays

Antibodies were immobilised either by plate binding or bead-binding. For plate binding, antibodies were diluted to 10 μg/mL in 0.1M $NaHCO_3$ and incubated overnight at 4°C, 500 μl per well of a 24-well plate. The next day, the antibody was aspirated and wells were washed three times with PBS before being incubated with cells in complete IMDM media and any soluble antibodies. For bead-binding, 10 μg of biotinylated antibodies were incubated with 100 μl of EasySep Biotin selection cocktail (EasySep Biotin positive selection kit II, STEMCELL) for 15 minutes at room temperature. 50 μl of EasySep Dextran RapidSpheres were then added to the mixture and further incubated for 10 minutes at room temperature. Following incubation, the antibody-bead complexes were purified by two rounds of washing and magnetic separation. Prepared antibody-bead complexes were used to set up cellular assay immediately.

### Antibodies

All anti-envelope antibodies and isotype control antibodies were in-house purified by affinity chromatography (low-endotoxin, azide-free): endogenous MLV envelope was detected using the 83A25 antibody (rat IgG2a) [51], F-MLV envelope was detected using the 720 antibody (mouse IgG1) [57], and xenotropic MLV envelope was detected using the 522 antibody (mouse IgM) [56]. Phospho-p44/42 MAPK (Erk1/2) (Thr202/Tyr204) (clone D13.14.4E), phospho-CREB (Ser133) (clone87G3), pan-actin (D18C11) and rabbit (DA1E) isotype control antibodies were from Cell Signaling Technology. Goat anti-Rabbit IgG (H+L) Cross-Adsorbed, Alexa Fluor 488 (A-11008) secondary antibody was from Thermo Fisher Scientific. All other antibodies were from Biolegend: LEAF-purified anti-mouse CD11a (clone M17/4), LEAF-purified anti-mouse CD3e (clone 145-2C11), purified anti-mouse CD5 (clone 53–7.3), anti-mouse CD45 Brilliant Violet 570 (clone 30-F11), anti-mouse CD43 PE (clone S11), anti-mouse CD19 PE (clone, 6D5), anti-mouse CD3 FITC (clone 145-2C11), anti-human CD45

Brilliant Violet 570 (clone HI30), anti-mouse CD107a PE (clone 1D4B). Fc blocking antibody, anti-mouse CD16/CD32 (clone 2.4G2) was from BD Biosciences.

## Antibody conjugation

Antibodies were conjugated to Alexa Fluor 647 using Antibody Labeling Kit (ThermoFisher), pHrodo iFL Green using Protein labelling kit (ThermoFisher) or Biotin using EZ-Link Micro Sulfo-NHS-Biotinylation kit (ThermoFisher) according to manufacturer's instructions.

## Flow cytometry

Single-cell suspensions from spleens were prepared by mechanically disrupting the spleens through 70 μm nylon filters. Red blood cells were lysed by incubating the suspensions in 0.83% Ammonium Chloride for 5 minutes at room temperature. The cells were then washed and resuspended in PBS / 2% FCS. Staining with antibodies was performed for 30 minutes at room temperature. For intracellular staining, cells were fixed for 15 minutes in 4% paraformaldehyde, washed with PBS / 2% FCS and permeabilised by adding ice-cold 90% methanol and incubating on ice for 30 minutes. Cells were then washed twice with PBS / 2% FCS and incubated with primary antibodies diluted in PBS / 2% FCS for 1 hour at room temperature. Cells were washed twice with PBS / 2% FCS and incubated with the secondary antibody diluted in PBS / 2% FCS for 30 minutes at room temperature. Cells were then washed twice in PBS / 2% FCS and resuspended in PBS / 2% FCS for analysis. Flow cytometric data was collected on a Fortessa cell analyser (Becton Dickinson) and analysed using FlowJo 10 software.

## ImageStream

Cells were cultured in complete media with internalising antibody at 10 μg/mL for specified periods of time at 37˚C in a 5% $CO_2$ incubator. Counter staining antibodies were added 30 minutes prior to analysis followed by Hoechst 33342 nuclear stain (Thermo Fisher) at 0.2 μg/ mL at 10 minutes prior to analysis. Acidic organelles were stained using LysoTracker Red (ThermoFisher) at 17 nM according to manufacturer's instructions. Cells were washed and resuspended in PBS / 2% FCS with 20 ng/mL propidium iodide. Samples were acquired on a 5-laser 12-channel Amnis ImageStream Mk II Imaging Flow Cytometer at 60 × magnification controlled by INSPIRE software and fully ASSIST calibrated. Single colour controls were acquired in order to calculate the fluorochrome-specific compensation matrix, which was then applied to all of the sample image files to achieve spectral un-mixing. Internalisation and co-localisation analysis were performed using IDEAS 6.2 Internalisation and Co-localisation wizards. In short, cells were gated on Focused, Single cells, Live cells and positive for any surface or nuclear markers used. Percentage of cells with internalised antigen was calculated using CD45 surface stain as a marker of cellular membrane (except G7 cell line where brightfield image was used as a cell image).

## Western blotting

Cells were lysed in SDS sample buffer and denatured for 15 minutes at 95˚C. Samples were analysed by standard immunoblotting techniques [78].

## Gene synthesis, cloning and mutagenesis

*Env* genes were synthesised and cloned into pRV-IRES-GFP vector upstream of IRES sequence. Mutations were introduced using QuickChange site-directed mutagenesis kit (Agilent). Deletion of the cytoplasmic tail of Emv2 envelope was achieved by introducing a STOP

codon at position 445–447 (AAT to TAA mutation) to create the Emv2 envelope DC mutant. Deletion of Emv2 envelope internalisation signal was achieved by deleting TATCATCAACT-TAA sequence at position 511–525 to create the Emv2 envelope DIS mutant. Emv2 envelope Y656A mutant was created by replacing TAT with GCT at position 511–513. All gene synthesis, cloning and mutagenesis were performed by Genewiz LLC. All of the resulting constructs were verified by sequencing.

## Retroviral transduction

Wild-type and mutant *env* genes were introduced into cell lines by means of retroviral transduction. Retroviral particles carrying the *env* genes were generated by co-transfecting HEK293T cells with pHIT60 (encoding gag/pol), pVSVG (encoding VSVG glycoprotein) and pRV-IRES-GFP carrying the desired *env* gene. Transfection was performed using GeneJuice transfection reagent (Novagen) according to manufacturer's instructions. 48 hours following transfection, viral supernatant was collected, filtered through a 0.45 μm filter and used to transduce cell lines. Transduction was achieved by centrifugation of cell lines with viral supernatant in the presence of polybrene (4 μg/mL) at 300 *g* for 45 minutes at room temperature. 72 hours post transduction, cells expressing the *env* gene were selected by cell sorting using GFP and envelope staining as markers for positive selection. All cell lines had undergone at least three rounds of selection. Cell sorting was performed using FACSAria Fusion cell sorter (Becton Dickinson).

## Expression analyses by RNA-seq

Cellular RNA was extracted, treated with DNaseI and sequenced with PE100 reads on Illumina HiSEQ 2500. Gene and endogenous retroelement expression was assessed as previously described [49]. Data were deposited at the EMBL-EBI repository (www.ebi.ac.uk/arrayexpress) under accession numbers ERP120018 (EL4 cells) and ERP120023 (Jurkat cells). Hierarchical clustering and heatmap production was with Qlucore Omics Explorer (Qlucore, Lund, Sweden). Pathway analyses were performed using g:Profiler (https://biit.cs.ut.ee/gprofiler).

## Confocal microscopy

Cells were fixed in 4% paraformaldehyde for 15 minutes on ice before being permeabilised in PBS / 3% BSA / 0.05% Triton for 20 minutes at room temperature. Cells were then incubated with unconjugated 83A25 primary antibody diluted in PBS / 3% BSA / 0.05% Triton for 1 hour at room temperature. Following three rounds of washing with PBS / 0.1% BSA / 0.05% Triton, the cells were incubated with anti-rat Alexa Fluor 488 (A-11006, ThermoFisher) secondary antibody diluted in PBS / 3% BSA / 0.05% Triton for 1 hour at room temperature. Cells were washed three times with PBS / 0.1% BSA / 0.05% Triton before being counter stained with DAPI and mounted using Vectashield mounting medium. Images were acquired using Leica SP5 or Zeiss LSM 880 confocal microscopes with plan apochromat 63 ×, NA 1.40 objectives. The images were analysed using ImageJ software (NIH).

## Light microscopy

Live cells were imaged using EVOS FL imaging system (ThermoFisher) fitted with a LPlanFL PH2 10 ×, NA 0.3 objective. Quantitative image analysis was performed using ImageJ software (NIH).

## PCR and quantitative RT-PCR

Genomic DNA from mouse tail biopsies was extracted by incubating the samples overnight in the following digestion buffer: 100 mM Tris, 5 mM EDTA, 200 mM NaCl, 0.2% SDS, and 0.4 mg/mL Proteinase K. Following incubation, the samples were centrifuged at 16 000 × g for 10 minutes, and the supernatants were transferred into clean tubes containing one volume of iso-propanol. DNA was pelleted by centrifugation, washed with ethanol, air-dried, and resuspended in $dH_2O$. PCRs were set up using Phusion High-Fidelity PCR Kit (ThermoFisher).

Cellular RNA was extracted using the RNeasy Mini QIAcube Kit (QIAGEN), treated with DNaseI (QIAGEN) and reverse transcribed into cDNA using High-Capacity Reverse Transcription Kit (ThermoFisher). The reactions were purified using QIAquick PCR purification kit (QIAGEN). Q-PCRs were set up using self-designed primers and Fast Sybr Green Master Mix (ThermoFisher), and run on QuantStudio 5 Real Time PCR System (ThermoFisher). Gene expression was normalised to either mouse or human HPRT expression. The comparative $C_T$ method ($\Delta\Delta C_T$) was used to analyse the resulting data for the relative quantitation of gene expression.

Sequences of primers used are shown in S1 Table.

## Statistical analyses

All statistical analyses were performed using Prism 7 software (GraphPad). Results are shown as mean ±SEM. Parametric comparisons of normally distributed values that satisfied the variance criteria were made by unpaired Student's t-tests or One Way Analysis of variance (ANOVA) tests. Data that did not pass the variance test were compared with non-parametric two-tailed Mann-Whitney Rank Sum tests or ANOVA on Ranks tests. For cellular assays, each treatment group was compared to the control group. Multi-group comparisons (F-test) on transcriptomic data were run by ANOVA in Qlucore Omics Explorer. Correction for multiple tests was achieved with the use of q-value cut-off of 0.05 (based on the False Discovery rate). Results were considered significant at $^*P \leq 0.05$; $^{**}P \leq 0.01$; $^{***}P \leq 0.001$, $^{****}P \leq 0.0001$.

## Supporting information

**S1 Fig. PCR analysis of the ancestral locus of *Xmv45* integration.** (A) Schematic representation of reconstructed locus prior to *Xmv45* integration, referred to as *AI506816a*, depicting the position of PCR primers used. (B) PCR results from genomic DNA from the indicated inbred mouse strains.
(TIF)

**S2 Fig. MLV envelope expression and distribution on the plasma membrane.** (A) Representative confocal images showing distribution of MLV envelope proteins from endogenous retroviruses in 2695 B cell lymphoma line. Cells were fixed, permeabilised and labelled with 83A25. Scale bar = 5 μm. (B) Flow cytometric analysis of Emv2 envelope expression on the surface of Jurkat.Emv2env cells (left) and representative confocal images showing distribution of Emv2 envelope in transduced Jurkat cells (right). Cells were fixed, permeabilised and labelled with 83A25. Scale bar = 5 μm.
(TIF)

**S3 Fig. MLV envelope is internalised in multiple tumour cell lines.** IS images of various tumour cell lines incubated with 83A25 for 3 hours and counterstained with anti-CD45 and Hoechst (left). Quantification of cells with internalised envelope-antibody complexes (right). A minimum of 5000 cells were analysed at each time point. Scale bar = 7 μm.
(TIF)

**S4 Fig. Endogenous and exogenous MLV envelopes are similarly internalised.** (A) Xenotropic envelope is internalised within three hours of incubation with the 522 antibody. IS images of E.G7-OVA cells incubated with 522 for specified periods of time and counterstained with anti-CD45 and Hoechst (left). Quantification of cells with internalised envelope/antibody complexes (right) from two independent experiments. A minimum of 10000 cells were analysed in each experiment at each time point. Scale bar = 7 μm. (B) F-MLV envelope is internalised within three hours of incubation with the 720 antibody. F-MLV FB29env-transduced EL4 cells were incubated with 720 for specified periods of time, counterstained with anti-CD45 and Hoechst and imaged by IS (left). Flow cytometric analysis of F-MLV FB29 envelope expression on the surface of EL4.FB29env cells (top right). Quantification of cells with internalised F-MLV FB29 envelope-antibody complexes (bottom right) from two independent experiments. A minimum of 10000 cells were analysed in each experiment at each time point. Scale bar = 7 μm. (C) Emv2 envelope is internalised within 3 hours of incubation with the 83A25 antibody. Emv2env-transduced Jurkat cells were incubated with 83A25 for specified periods of time, counterstained with anti-CD45 and Hoechst and imaged by IS (left). Quantification of cells with internalised Emv2 envelope-antibody complexes (right) from two independent experiments. A minimum of 10000 cells were analysed in each experiment at each time point. Scale bar = 7 μm.
(TIF)

**S5 Fig. CD45 is not internalised by EL4 and Jurkat cells.** (A) IS images of EL4 cells incubated with anti-mouse CD45 for indicated periods of time and counterstained with Hoechst (left). Quantification of cells with internalised CD45-antibody complexes compared to envelope-antibody complexes (right). A minimum of 10000 cells were analysed at each time point. Scale bar = 7 μm. (B) IS images of Jurkat cells incubated with anti-human CD45 antibody for indicated periods of time and counterstained with Hoechst (left). Quantification of cells with internalised CD45-antibody complexes compared to envelope-antibody complexes (right). A minimum of 10000 cells were analysed at each time point. Scale bar = 7 μm.
(TIF)

**S6 Fig. Stimulation of A1 cells with 83A25 induces cell clustering.** Light microscopy images of A1 cells incubated with indicated antibodies for 18 hours (left). Scale bar = 200 μm. Quantification of area occupied by the cells as a percentage of total per field of view (right). Pooled data from three independent experiments with at least ten fields of view per experiment.
(TIF)

**S7 Fig. Transcriptional activation induced by MLV envelope ligation.** Changes in gene expression in EL4 cells over time following incubation with 83A25, isotype control antibody or CD3 and CD28 Dynabeads. Each column is an independent replicate.
(TIF)

**S8 Fig. Verification of differentially expressed genes by qRT-PCR analysis.** Expression of *Rab30*, *Fam129b*, *Ahnak*, *Ggt1* and *Thbs2* genes assessed by qRT-PCR in EL4 cells stimulated with 83A25 for 18 hours.
(TIF)

**S9 Fig. Transcriptional activation following envelope ligation by antibody or cellular receptor, but not by Fc receptor ligation.** (A) Expression of *Nhsl2* and *Tgfb3* genes assessed by qRT-PCR in EL4 cells stimulated with 83A25 for 18 hours in the presence of 2.4G2 FcR blocking antibody. (B) Expression of *Nhsl2* and *Tgfb3* genes assessed by qRT-PCR in EL4.

FB29env cells stimulated with the 720 antibody and in EL4 cells stimulated with the 522 antibody for 18 hours. Pooled data from two (522) and three (720) independent experiments. (C) Expression of *Nhsl2* and *Tgfb3* genes assessed by qRT-PCR in EL4 cells co-cultured with 293T or 293T.mCAT-1 cells for 18 hours. Pooled data from three independent experiments.
(TIF)

**S10 Fig. Dose-dependent effect of 83A25 antibody ligation of MLV envelope.** EL4 cells were stimulated with various concentrations of 83A25 for 18 hours and levels of *Nhsl2* and *Tgfb3* gene expression were assessed by qRT-PCR.
(TIF)

**S11 Fig. Colocalisation of internalised MLV envelope and CD5.** (A) CD5 is internalised into the same vesicles as 83A25-envelope complexes. IS images of EL4 cells co-incubated with 83A25 and anti-CD5 for specified periods of time and stained with Hoechst (top panel). Scale bar = 7 μm. Quantification of cells with internalised envelope-antibody complexes (bottom left). A minimum of 5000 cells were analysed at each time point. Co-localisation of 83A25 with CD5 was quantified using the Bright Detail Similarity feature in IDEAS and compared to Hoechst, a non-colocalising probe (bottom right). (B) Expression of *Nhsl2* and *Tgfb3* genes assessed by qRT-PCR in EL4 cells stimulated with anti-CD5 for 18 hours. Pooled data from two independent experiments.
(TIF)

**S12 Fig. Constitutive activation of ERK and CREB in EL4 cells.** (A) Flow cytometric analysis of intracellular phospho-ERK (pERK) and phospho-CREB (pCREB) in resting EL4 cells and following stimulation with the indicated antibodies for 20 minutes. Grey-filled histograms represent the isotype control for the staining. Data representative of three independent experiments. (B) Western blot analysis of pERK and pCREB in resting EL4 cells and following stimulation with the indicated antibodies for 20 minutes. Data representative of one experiment.
(TIF)

**S13 Fig. Transcriptional effects of MLV envelope in Jurkat.Emv2env cells.** Heatmap of differentially expressed genes ($\geq$2-fold, q$\leq$0.05) between Jurkat and Jurkat.Emv2env cells (left) and pathway analysis of these genes, according to g:Profiler (https://biit.cs.ut.ee/gprofiler).
(TIF)

**S14 Fig. Transcriptional activation is proportional to MLV envelope expression.** (A) *FLT3*, *NELL1* and *PRKG2* gene expression correlates with Emv2 envelope expression levels on the cell surface. Jurkat.Emv2env cells were sorted for Emv2 envelope low or high (top) and assessed for expression of *FLT3*, *NELL1* and *PRKG2* genes by qRT-PCR (bottom). (B) Verification of differentially expressed genes by qRT-PCR analysis. Expression of *FLT3*, *NELL1* and *PRKG2* genes in Jurkat.Emv2env and Jurkat.GFP cells assessed by qRT-PCR.
(TIF)

**S15 Fig. Cytoplasmic tail deletion diminishes envelope expression on the cell surface.** Flow cytometric analysis of Jurkat.Emv2env ΔCT cells for surface (left) and intracellular (right) expression of Emv2 envelope.
(TIF)

**S1 Table. Sequence of PCR primers used in this study.**
(PDF)

## Acknowledgments

Drs. Kim J. Hasenkrug, Bruce W. Chesebro and Leonard H. Evans for generously providing the 720, 522 and 83A25 monoclonal antibodies and Dr Walther Mothes for the 293T.mCAT-1 cells. We are grateful for assistance from the Flow Cytometry and Biological Resource Facilities at the Francis Crick Institute.

## Author Contributions

**Conceptualization:** Jonathan P. Stoye, George Kassiotis.

**Data curation:** Veera Panova, Jan Attig, George R. Young.

**Formal analysis:** Veera Panova, Jan Attig, George R. Young.

**Funding acquisition:** George Kassiotis.

**Investigation:** Veera Panova, Jan Attig, George R. Young.

**Supervision:** Jonathan P. Stoye, George Kassiotis.

**Visualization:** Veera Panova, George Kassiotis.

**Writing – original draft:** Veera Panova, Jan Attig, Jonathan P. Stoye, George Kassiotis.

**Writing – review & editing:** Veera Panova, Jan Attig, Jonathan P. Stoye, George Kassiotis.

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
