## [Decision Letter · Decision Letter 0]

5 Mar 2020

Dear Dr. Kassiotis,

Thank you very much for submitting your manuscript "Antibody-induced internalisation of retroviral envelope glycoproteins is a signal initiation event" for consideration at PLOS Pathogens. As with all papers reviewed by the journal, your manuscript was reviewed by members of the editorial board and by three independent reviewers. In light of the reviews (below this email), we would like to invite the resubmission of a significantly-revised version that takes into account the reviewers' comments. In particular, the reviewers felt that mechanistic insight into how envelope internalization affected cell signaling, as well as the biological significance of this, was lacking.

We cannot make any decision about publication until we have seen the revised manuscript and your response to the reviewers' comments. Your revised manuscript is also likely to be sent to reviewers for further evaluation.

Sincerely,

Susan Ross

Section Editor

PLOS Pathogens

Kasturi Haldar

Editor-in-Chief

PLOS Pathogens

orcid.org/0000-0001-5065-158X

Michael Malim

Editor-in-Chief

PLOS Pathogens

orcid.org/0000-0002-7699-2064

Reviewer's Responses to Questions

**Part I - Summary**

Reviewer #1: Previous work already showed that antibody binding to viral envelopes (RSV, FCoV and HIV) results in Ab-Env complex internalisation. In this manuscript, Panova and collaborators extend these observations to murine leukaemia viruses and observe a similar phenotype. However, they go beyond this original phenotype and describe that antibody binding to MLV Env induces significant signalling as measured by RNAseq and confirmed by qRT-PCR. Importantly, they show that Ab-induced internalisation is required for signalling. Moreover, they show that under some circumstances signalling translate into cell activation. This new information might have implications for viral eradication strategies but also cancer therapies. Overall, this is a well-written and balanced manuscript. The experiments are technically sound, well controlled and certainly well executed. The substantial amount of results is of high quality. This refreshing manuscript brings new and interesting information to the field.

Reviewer #2: This is a well-written paper reporting the results of relatively well-planned and carefully performed experiments. All data are presented well in convincing figures. However, biological significance of the reported observations has not been thoroughly pursued. The authors themselves admit in the middle of page 17 that a possible mechanistic link between envelope-induced signaling in T cells and induction of leukemia/lymphoma in vivo would be worth exploring. Such pathophysiological analyses are exactly what readers of this journal expect from this group of authors.

Reviewer #3: In the manuscript entitled “Antibody-induced internalization of retroviral envelope glycoproteins is a signal initiation event,” Panova and colleagues show that envelope protein derived from endogenous murine leukemia virus (MLV) is internalized from the plasma membrane into endocytic compartments upon engagement with anti-envelope antibody in mouse T and B cells. The authors showed that this is independent of Fc and leads to a range of signaling events including cell clustering, degranulation, and transcriptional changes. Finally, the authors demonstrate that transcriptional changes require the YXXΦ motif in the cytoplasmic tail of the envelope protein.

The data suggest signaling from endogenous retrovirus envelope protein induced by anti-envelope antibodies and potentially host viral receptors. However, the mechanism by which internalization of the envelope protein results in changes to the cellular signaling pathways is largely absent. Furthermore, numerous areas throughout the manuscript contain inconsistencies and require clarification.

**Part II – Major Issues: Key Experiments Required for Acceptance**

Reviewer #1: 1- Previous studies have shown that antibody-mediated Env internalisation depends on their capacity to crosslink. Fab fragments appear to be unable to induce Env internalisation. Is this also true for MLV envelope glycoproteins? Does 83A25 Fab fragment induce internalisation?

2- Similarly, is antibody-mediated crosslinking required to signal? This could be easily tested in several ways, for example by measuring by qRT-PCR the upregulation of Nhsl2 RNA (Figure 3F) induced by 83A25 compared to its Fab fragment.

3- Is the transcriptional activation and signalling unique to 83A25 or also observed with 522 and 720? (these antibodies also induce Env internalisation).

Reviewer #2: 1) Most of presented data are obtained by using cultured cell lines and transfectants. While it is clear that the ligation of cell surface envelope molecule either with an antibody or with mCAT-1 receptor induces envelope protein internalization and cytoplasmic YXXphai motif is required for this process, and the above envelope ligation results in the induction of a battery of genes in EL4 and transfected Jurkut cells that are different from those induced by CD3/CD28 stimulation, it is unclear if this unique signaling is associated with any particular type of cell activation in primary T and B lymphocytes or other cells of the immune system.

2) In this regard, while the authors describe that MLV envelope glycoprotein expression is linked with immune activation and antibody responses elicited against endogenous retroviral envelope proteins in the Abstract and Author Summary, they have failed to cite and discuss about important previous reports that are highly relevant to the relationship between immune activation and antibody production against endogenous retroviral envelope glycoproteins. John Portis and his colleagues isolated a large number of monoclonal antibodies reactive with endogenous xenotropic and polytropic viral envelope glycoproteins by inducing graft-versus-host reaction (Portis, J. L., F. J. McAtee, M. W. Cloyd. Virology 118:181-190, 1982; Portis, J. L. and F. J. McAtee. Virology 126:96-105, 1983). Generation of a large number of hybridomas reacting to endogenous retroviral rather than host antigens was unexpected, and he later discussed the possibility of antigen-driven activation of autoreactive lymphocytes (Portis, J. L. Tohoku J. Exp. Med. 173:83-89, 1994). The authors should cite these papers and discuss the possible relationships between their observation and the induction of antiretroviral antibodies upon immune activation.

3) In Figure 1, the authors first examined the expression of endogenous retroviruses in unstimulated and stimulated CD4+ T cells and then abruptly switch to LPS stimulation of B cells. While the genetic analyses on the mechanisms of LPS responsiveness are very important, is there any rationale to start with CD4+ T cells, switch to LPS stimulation of B cells, and come back to EL-4 T cell line?

4) In this regard, the expression of endogenous xenotropic viral envelope gene(s) and the production of "autoantibodies" reactive with the envelope glycoprotein have long been associated with the pathogenesis in lupus-prone mice as the authors briefly mention, and a lack of association of antinuclear antibodies with glomerulonephritis or vasculitis (Christensen, S. R. et al. Immunity 25:417-428, 2006) and direct pathogenicity of autoantibodies reactive with "xenotropic" viral gp70 have been reported (Tabata, N. et al. J. Virol. 74:4116-4126, 2000). Of particular importance, retroviral gp70-containing circulating immune complexes are associated with the development of glomerulonephritis in murine models of SLE and the source of serum gp70 is the liver where gp70 is expressed as an acute phase reactant (Shigemoto, K. et al. Mol. Immunol. 29:573-582, 1992). Thus, there must be some analyses and discussion on the possible relationship between the LPS-responsive Xmv45 and the gene encoding the serum gp70 that is expressed as acute phase reactant.

Reviewer #3: 1. In Figure 1, the authors show that Xmv45 is highly expressed in CD4 T cells but describe a chimeric transcript of Xmv45-AI506816a that is induced in LPS-activated B cells. What is the relevance of this finding? Is this chimeric transcript also expressed in activated T cells? Based on RNA sequencing, primary CD4+ T cells have decreased expression of Xmv45 upon CD3/CD28 stimulation (Figure 1a). How does this translate to surface expression of envelope in all of these cell types? Does CD3/CD28 stimulation also induce Xmv45 expression in EL4 cells, which are used in the majority of the experiments?

2. The authors claim that MLV envelope induction upon LPS stimulation is absent in 129S8 and BALB/c (Figure 1D) but the data lack unstimulated controls for each strain, thus the induction cannot be assessed.

3. As demonstrated in Figure 2B, CD3 and CD19 are similarly endocytosed in primary lymphocytes upon 83A25 engagement (Figure 2B). These data suggest that the downstream signaling effects of 83A25-dependent envelope internalization described in EL4 cells (Figure 3) are potentially confounded by signaling through the internalized CD3. The authors should determine whether downstream consequences of envelope internalization (clustering, degranulation, transcriptional changes) are independent of CD3 internalization in EL4 cells.

4. The authors show that CD107a expression by EL4 cells stimulated with 83A25 is comparable to engagement of the envelope protein by the cognate receptor mCAT1. The authors should show whether mCAT1-dependent CD107a expression requires the same internalization process as 83A25 stimulation, and if not, explain the mechanism of internalization versus receptor engagement on downstream signaling.

5. The authors show that Nhsl2 and Tgfb3 induction upon 83A25 stimulation of EL4 cells is dependent on protein kinase A, Abl kinase, and Src kinase using chemical inhibitors. These results would be strengthened by demonstration that these pathways are indeed activated upon 83A25 engagement.

6. Based on experiments in mouse cells, the authors conclude that antibody-mediated internalization of the envelope protein from the plasma membrane to the endocytic compartment results in activation of cellular signaling. However, in human Jurkat cells, the authors show that transcriptional changes requires the sorting motif within the cytoplasmic tail, even though internalization of the envelope protein is only partially affected by this motif. These data seem to decouple internalization of envelope from transcriptional changes and contradict their main conclusion.

**Part III – Minor Issues: Editorial and Data Presentation Modifications**

Reviewer #1: The authors suggest (in the abstract) that retroviral Env signalling could be a target for intervention. It is unclear how this would be achieved. For example, anti-Env Abs appear to induce signalling. Which intervention do the authors have in mind? Small molecules? Genetic approaches? Consider discussing further this intriguing idea.

Antibody-induced Env internalisation is not new and was previously-described for several viruses. I was surprised to see that this was only mentioned in the discussion; completely absent from the introduction section of the manuscript.

A better description on how the % of internalisation was calculated will facilitate the understanding of all the internalisation graphs presented in the manuscript.

On page 10 (result section describing Figure 3E), the authors should explain the rationale behind focusing only on Nhsl2, Tgfb3, Abhd12b and Pmaip1 out of all the genes that are upregulated upon 83A35 stimulation. What do these genes transcribe for and what is their relevance in signalling pathways as previously described in the literature? Similarly, the rationale for studying only Nhsl2 and Tgfb3 transcription in their antibody-immobilization assay and signalling inhibition assays in Figure 4.

Reviewer #2: 1) In Figure 1 legend, an "immune response gene" means a host gene that regulate immune responsiveness like those in the major histocompatibility complex. The authors here are referring to a gene that is responsive to immunologic stimulations, which is not described as immune response gene.

2) In Figures 3, 4 and 5, statistical comparisons are made between multiple groups but the Materials and Methods describes only the use of one-way ANOVA and it is unclear if any post-hoc tests were performed for multiple comparisons. Further, while multiple asterisks are shown probably to indicate different levels of statistical significance, no explanations are provided in corresponding legends.

3) In Figure 5, panel C, transfection efficiency was estimated by cotransfection of GFP-expressing plasmid, and thus expression levels of Emv2env were not directly measured. The labeling of Emv2env neg, int and high are misleading, unless the authors directly tested the envelope expression levels with a specific antibody.

Editorial issues:

a) In the last line on page 2, "envelops glycoproteins" must be amended.

b) In Figure 5 legend for panel C, "a GFP only carrying virus" must be rephrased.

Reviewer #3: 1. What is the rationale for overexpressing Emv2 envelope in Jurkat cells instead of Xmv45 envelope?

2. On page 12, the authors state, “RNA-seq analysis over the course of stimulation revealed upregulation of EGR1 and PTCRA and downregulation of RAG1, RAG2, and DLX3 in Jurkat.Emv2env cells stimulated with 83A25 or CD3/CD28, but not in unstimulated cells.” However, the data do not show elevated PTCRA in either 83A25 or CD3/CD28 stimulated cells, instead it shows no difference at 3hrs and reduced expression at 18hrs for 83A25-treated cells and at both time points for CD3/CD28-stimulated cells. Similarly, although DLX3 is reduced in transduced cells in the absence of stimulation, it is induced upon both 83A25 and CD3/CD28 stimulation.

3. In the discussion, the authors compare their study to antibody-induced internalization of the HIV-envelope. However, the referenced HIV study by Anand et al. (JVI 2019) is based on antibody-dependent cellular cytotoxicity, which is a FcR-dependent process and is therefore not comparable to the existing study given the Fc-independent process described in this manuscript.

PLOS authors have the option to publish the peer review history of their article (what does this mean?). If published, this will include your full peer review and any attached files.

Reviewer #1: No

Reviewer #2: No

Reviewer #3: No
---

## [Editor Report · Decision Letter 1]

5 May 2020

Dear Dr. Kassiotis,

We are pleased to inform you that your manuscript 'Antibody-induced internalisation of retroviral envelope glycoproteins is a signal initiation event' has been provisionally accepted for publication in PLOS Pathogens.

Best regards,

Susan R. Ross, PhD

Section Editor

PLOS Pathogens

Susan Ross

Section Editor

PLOS Pathogens

Kasturi Haldar

Editor-in-Chief

PLOS Pathogens

orcid.org/0000-0001-5065-158X

Michael Malim

Editor-in-Chief

PLOS Pathogens

orcid.org/0000-0002-7699-2064
---

## [Editor Report · Acceptance letter]

19 May 2020

Dear Dr. Kassiotis,

We are delighted to inform you that your manuscript, "Antibody-induced internalisation of retroviral envelope glycoproteins is a signal initiation event," has been formally accepted for publication in PLOS Pathogens.

Best regards,

Kasturi Haldar

Editor-in-Chief

PLOS Pathogens

orcid.org/0000-0001-5065-158X

Michael Malim

Editor-in-Chief

PLOS Pathogens

orcid.org/0000-0002-7699-2064